Assessment of North American arthropod collections: prospects and challenges for addressing biodiversity research

Cobb Neil S. neil.cobb@nau.edu 1
Gall Lawrence F. 2
Zaspel Jennifer M. 3 4
Dowdy Nicolas J. 3 5
McCabe Lindsie M. 1
Kawahara Akito Y. 6
1 Department of Biological Sciences, Northern Arizona University , Flagstaff , AZ , United States of America
2 Entomology Division, Yale Peabody Museum of Natural History , New Haven , CT , United States of America
3 Department of Zoology, Milwaukee Public Museum , Milwaukee , WI , United States of America
4 Department of Entomology, Purdue University , West Lafayette , IN , United States of America
5 Department of Biology, Wake Forest University , Winston-Salem , NC , United States of America
6 Florida Museum of Natural History, University of Florida , Gainesville , FL , United States of America
Bieler Rudiger
Electronic publication date: 2019 Nov 25
Publication date: 2019
Volume: 7
Electronic Location ID: e8086
Received 2019 Aug 2; Accepted 2019 Oct 23
Copyright: ©2019 Cobb et al.
Copyright year: 2019
Copyright holder: Cobb et al.
License: This is an open access article distributed under the terms of the Creative Commons Attribution License, which permits unrestricted use, distribution, reproduction and adaptation in any medium and for any purpose provided that it is properly attributed. For attribution, the original author(s), title, publication source (PeerJ) and either DOI or URL of the article must be cited.
License URL: https://creativecommons.org/licenses/by/4.0/

Keywords: Arthropods, Natural history collections, Biodiversity, North America, Digitization

Funding: National Science Foundation EF 1207371 DBI 1602081 DBI 1759966 DBI 1600616 DBI 1561448 DBI 1601957 DBI 1811897 DBI 1601957 DBI 1601369 The following National Science Foundation grants comprise all the external funding or sources of support received during this study: EF 1207371, DBI 1602081, DBI 1759966 to Neil S Cobb; DBI 1600616 to Lawrence F Gall; DBI 1561448, DBI 1601957 to Jennifer M Zaspel; DBI 1811897 to Nicolas J Dowdy; and DBI 1601369 to Akito Y Kawahara. There was no additional external funding received for this study. The funders had no role in study design, data collection and analysis, decision to publish, or preparation of the manuscript.

==============================
Over 300 million arthropod specimens are housed in North American natural history collections. These collections represent a “vast hidden treasure trove” of biodiversity −95% of the specimen label data have yet to be transcribed for research, and less than 2% of the specimens have been imaged. Specimen labels contain crucial information to determine species distributions over time and are essential for understanding patterns of ecology and evolution, which will help assess the growing biodiversity crisis driven by global change impacts. Specimen images offer indispensable insight and data for analyses of traits, and ecological and phylogenetic patterns of biodiversity. Here, we review North American arthropod collections using two key metrics, specimen holdings and digitization efforts, to assess the potential for collections to provide needed biodiversity data. We include data from 223 arthropod collections in North America, with an emphasis on the United States. Our specific findings are as follows: (1) The majority of North American natural history collections (88%) and specimens (89%) are located in the United States. Canada has comparable holdings to the United States relative to its estimated biodiversity. Mexico has made the furthest progress in terms of digitization, but its specimen holdings should be increased to reflect the estimated higher Mexican arthropod diversity. The proportion of North American collections that has been digitized, and the number of digital records available per species, are both much lower for arthropods when compared to chordates and plants. (2) The National Science Foundation’s decade-long ADBC program (Advancing Digitization of Biological Collections) has been transformational in promoting arthropod digitization. However, even if this program became permanent, at current rates, by the year 2050 only 38% of the existing arthropod specimens would be digitized, and less than 1% would have associated digital images. (3) The number of specimens in collections has increased by approximately 1% per year over the past 30 years. We propose that this rate of increase is insufficient to provide enough data to address biodiversity research needs, and that arthropod collections should aim to triple their rate of new specimen acquisition. (4) The collections we surveyed in the United States vary broadly in a number of indicators. Collectively, there is depth and breadth, with smaller collections providing regional depth and larger collections providing greater global coverage. (5) Increased coordination across museums is needed for digitization efforts to target taxa for research and conservation goals and address long-term data needs. Two key recommendations emerge: collections should significantly increase both their specimen holdings and their digitization efforts to empower continental and global biodiversity data pipelines, and stimulate downstream research.

Introduction

Arthropod natural history collections

With more than one million described species, Arthropoda is the most taxonomically and ecologically diverse animal phylum, comprising over half of both North American and global animal species diversity (Briggs, 1994; Losey & Vaughan, 2006; Scudder, 2009; Stork, 2018). Arthropods include insects, arachnids, and crustaceans. Insects and arachnids are pervasive in non-marine environments and crustaceans dominate most marine environments. Arthropods are fundamental to ecosystem function and impact humans both positively and negatively (McIntyre, 2000). Arthropods are declining rapidly due to recent anthropogenic disturbances, such as climate change and various forms of pollution (Calvo-Agudo et al., 2019; Janzen & Hallwachs, 2019; Lister & Garcia, 2018; Sánchez-Bayo & Wyckhuys, 2019), underscoring an urgency in documenting their life histories and geographic distributions and preserving specimens for future research.

Here we examine 223 collections of arthropods in North America (Canada, Mexico and United States, including territories) that vary in size, governance, and locality (Fig. 1). Our overarching objectives include characterizing different types of arthropod collections, articulating challenges specific to arthropod collections, and assessing digitization efforts to date with a focus on meeting research data needs. We conducted analyses to examine broad scale trends concerning holdings and digitization efforts for all three countries but emphasize the United States (US) because we have the most complete data for that region. Collections assessed ranged from specialized small collections (∼500 specimens) to the United States National Museum of Natural History collection with 35 million specimens. Most of the North American collections have dedicated websites and are housed in universities, public museums, and repositories for government programs.

Figure 1 Map of North America showing the location of the arthropod collections included in the present study.

Location of all known arthropod collections in North America and US territories. Alaska and Hawaii are shown as inserts in lower left (Guam not shown). Map created using R packages rworldmap and ggmaps.

Our focus is on arthropod collections, which are dominated by insects (i.e., 96% of arthropod records discussed herein are for insects). At least 40% of North American insect collections include additional arthropod groups including Arachnida, Chilopoda, Crustacea, Diplopoda, and Entognatha (SCAN, 2019). A number of collections include invertebrates sensu lato, but we only surveyed those if they included insects. Additionally, we did not attempt to enumerate parasitic arthropods held in vertebrate collections (typically curated as data associated with vertebrate host specimens).

We also summarize digitization efforts across different collection sizes. Most small entomology collections are located within college and university departments, where the person responsible is a faculty member with a variety of additional responsibilities. These collections are often (1) focused on local fauna and/or reflect the particular interests of the curator(s), (2) managed and curated at their discretion, (3) lacking in dedicated institutional IT support, and (4) possibly supported by nominal budgets and/or students who receive credit for their participation. Larger arthropod collections are usually housed in museums that are either free-standing institutions or institutions affiliated with a larger university. These collections are typically (1) of regional or worldwide scope, (2) managed by a dedicated curator and/or collection manager, (3) have access to institutional IT support, and (4) are supported by longer-term budget commitments and access to institutional personnel and related resources. Although the potential capacity to produce digital products at larger collections is much greater than at small collections, the former are also embedded within a broader administrative infrastructure that often present other challenges.

Defining digitization for arthropod collections

Digitization is a term whose definition has been expanding in scope as technology allows more extraction of data from specimens (Nelson & Ellis, 2018; Short, Dikow & Moreau, 2018; Watanabe, 2019). We define digitization in the context of arthropod specimens as encompassing: (1) transcription of specimen labels into a database; (2) georeferencing localities (determining latitude/longitude); (3) capturing habitus image(s); and (4) vetting species-level identifications. These four elements of digitization are required to make records useful for most research purposes. Current digitization efforts focus almost exclusively on transcribing label data from specimens and georeferencing associated locality information, although some efforts have included historical field notes (Nufio et al., 2010). Most collections capture habitus images for exemplar specimens, but less than 1% of specimens have had a general habitus image recorded. Even fewer specimens have associated genetic data. There are some examples of collections linking genetic data to specimens (Short, Dikow & Moreau, 2018), or molecular tissue vouchers to specimens (Cho et al., 2016), but there is still rudimentary linkage between most genetic data in the Barcode of Life Datasystems (Ratnasingham & Hebert, 2007) and in similar genomic repositories and specimen occurrence databases.

To achieve the highest value for scientific research, digitization should extract all possible information from specimens, i.e., the “extended specimen” (Thiers, Mabee & Monfils, 2019) including morphological, anatomical, molecular, and possibly even metabolomic data. As technology advances and becomes more accessible, our ability to obtain massive amounts of data from specimens will rapidly increase. For example, recent studies have captured phenotypic trait data from arthropod specimens to examine response to environmental change over time (Kharouba et al., 2018; McLean et al., 2016). Morphological traits in insects are also beginning to be assessed via automated workflows for 3D modelling derived from multi-angle imaging (Ströbel et al., 2018) as well as from microCT data (Van de Kamp et al., 2015).

There are coarser levels of digitization that do improve the curation of holdings, including unit-tray or drawer-level inventories that estimate number of individuals per taxonomic rank as well as their level of curation (McGinley, 1993). These type of assessments are valuable and allow for an effective strategy to digitize individual specimens as well as guide future additions to the collection.

Importance of specimen-based data for biodiversity research

In the past two decades, digitized specimen records have become an invaluable resource for biodiversity and conservation research (Ball-Damerow et al., 2019). Plant and vertebrate collections have spearheaded this effort (Bakker, 2017; Bebber et al., 2010; Besnard et al., 2014; Bieker & Martin, 2018; Braun & Wann, 2017; Cook et al., 2014; Davis et al., 2015; Greve et al., 2016; Guralnick & Constable, 2010; Hart et al., 2014; Primack & Gallinat, 2017; Schmitt et al., 2018; Willis et al., 2017). Other natural history collections have followed the lead of plants and vertebrates (Brooks et al., 2014; Lawson et al., 2018). Digitization is of benefit to collections by allowing them to share their holdings with larger audiences, and opening new avenues for large-scale research and public engagement (Ellwood et al., 2015; Ellwood et al., 2018; Nelson & Ellis, 2018; Rapacciuolo et al., 2017; Spear, Pauly & Kaiser, 2017). Digitization also promotes collaborations among collections and integrated data at regional (Belitz et al., 2018; Sikes et al., 2016) and continental scales (Seltmann et al., 2017; Weirauch et al., 2017). Coordinated efforts to digitize arthropod collections across the US has resulted in an influx of specimen-level data and high-resolution images to online repositories (e.g., Symbiota Collection of Arthropods Network [SCAN], Global Biodiversity Information Facility [GBIF]). This in turn offers great potential to address an array of environmental issues such as climate change, impacts of human land use, agricultural intensification and the spread of human and animal disease, and the role of arthropods in ecosystem services and crop/forest pest management (Belitz et al., 2018; Bell-Sakyi et al., 2018; Cook et al., 2014; Dunnum et al., 2017; Kharouba et al., 2018; Meineke et al., 2018). Specimen data are also emerging as critical pedagogical resources for science educators seeking to enhance teaching curricula and data literacy (Cook et al., 2014; Ellwood et al., 2019; Lacey et al., 2017; Monfils et al., 2017; Singer, Love & Page, 2018).

Recent reviews of arthropod natural history collections and emerging collections-based research have focused on different aspects of the importance of digitized specimens. Short, Dikow & Moreau (2018) examined entomology collections in the context of “big data” with a focus on linking genetic data to specimens and technological advances in imaging. Bell-Sakyi et al. (2018) highlighted the importance and relevance of parasitic arthropod collections in understanding biotic interactions between disease vectors and their hosts (Pak, Jacobs & Sakamoto, 2019). Kharouba et al. (2018) studied collections-based research addressing global change impacts, with examples relating to geographical distributions, phenology, phenotypic and genotypic traits. Other reviews have summarized the importance of collections in general, and raised concerns over their sustainability as fundamental providers of biodiversity data and the invaluable expertise of collection personnel, curators, and research associates for preparing data products to support convergent research (Krishtalka & Humphrey, 2000; Thiers, Mabee & Monfils, 2019; Watanabe, 2019).

For taxonomic groups other than arthropods that have been the focus of digitization efforts for some time, there are recent assessments of the efficacy of such efforts and the state of collections as it relates to producing relevant biodiversity data. For example, Singer, Love & Page (2018) reviewed the major fish collections in the United States, updating holdings and digitization work over the last 22 years since the previous review by Poss & Collette (1995). Sierwald et al. (2018) provided a 40-year update on the survey of mollusk collections in the US and Canada since the previous review by Solem (1975). Our paper offers a comparable assessment of North American arthropod collections and establishes a baseline reference for future studies on museum and other research collections.

Survey Methodology

We began identifying collections and institutions for this survey in 2014 using the online resource “The Insect and Spider Collections of The World Website” (Evenhuis & Samuelson, 2019). More than 90% of the institutions we surveyed acknowledged the presence of a collection on their website. For all collections, we used the estimate of holdings listed on the collection website and in a few cases we followed up with direct correspondence to confirm holding size. We were reasonably confident that holding size did not include specimens in lots or large uncurated samples. Our list was compared periodically with several other resources: (1) a compendium of collections maintained by Song (2019); (2) collections listed in the database provided by the global registry of biodiversity repositories (Schindel & Cook, 2018); and (3) collections that were established through the SCAN data portal at https://scan-bugs.org (SCAN is a dedicated biodiversity portal that serves as an intermediate aggregator of data from 185 North American data providers) (SCAN, 2019). Our final list included 223 collections from across North America.

For analysis of accumulated digital records, we restricted the survey to collections that have made their specimen data publicly available through SCAN, GBIF (https://www.gbif.org/) and/or iDigBio (Page et al., 2015). The SCAN data portal was queried on 22 October 2018 and on 24 January 2019, and results were cross-checked against both GBIF and iDigBio. The SCAN portal contained over 18 million records for North America during that three-month assessment period. We excluded 1.5 million records that represented observation-only or image-only records, and another 3 million records that had incomplete or unresolved taxonomic and/or locality data. This yielded a 13.4 million record sample, and we assumed error rates in species identifications and locality data did not differ appreciably among the collections that had contributed records. Data analyses were conducted using R scripts on a computing cluster at Northern Arizona University (http://nau.edu/hpc/).

For the United States collections, we placed each collection surveyed into one of four size classes that included all terrestrial and freshwater aquatic arthropod records. The four classes from smallest to largest are: Tier 4 (<100,000 specimens); Tier 3 (100,000 to <1,000,000 specimens); Tier 2 (1,000,000 to 3,000,000 specimens); and Tier 1 (over 3,000,000 specimens). For temporal analysis, we defined a “historical record” as one where the collecting date was prior to 1965.

Results

Scope of North American arthropod collections and digitization efforts

Our survey of 223 arthropod collections from North America revealed that these collections currently house slightly more than 300 million specimens (Table S1), which is approximately triple the 93 million plant specimens estimated to be housed in North American herbaria (data from Index Herbariorum, March 2019, http://sweetgum.nybg.org/science/ih/). We were unable to determine an accurate estimate of the number of chordate (primarily vertebrates) specimens currently housed in North American collections, but that number is certainly smaller than for either plants or arthropods. These collection numbers do not strictly account for “specimen lots,” where multiple individual specimens are collected and preserved together. This is routine practice for arthropods but less common for chordates and plants. Most of our data are for single dry-preserved specimens representing lots of n = 1, and exclude immature arthropods, bulk samples, and other material typically stored in fluid or on slides as lots of n>1 (Sierwald et al., 2018). If we had been able to account for specimen lots, we believe the total number of arthropod specimens in North America would exceed 1 billion specimens (N Cobb, Derek Sikes, pers. comm., 2019). The overall pattern of records and diversity shows that compared to plants and especially vertebrates, arthropod records are much lower for North America compared to their diversity (Table 1).

Table 1 Metrics for North American collections for arthropoda, chordata, and plantae.

Species richness for Chordata estimated from (Dunnum, McLean & Dowler, 2018), for Plantae from (Ulloa et al., 2017) and for Arthropoda from Stork (2018). Data obtained from GBIF in January 2019.

	Arthropoda	Chordata	Plantae	
# Species	142,800	4,424	34,109	
# Specimen Records	13,788,159	11,430,528	13,787,883	
# Non-Specimen Records	3,335,975	329,994,473	6,729,368	
# Records/Species (Specimen Records)	97	2,584	404	
# Records/Species (Non-Specimen records)	23	74,597	197	

Table 1 presents summary statistics for digitization and species diversity for North American arthropod, plant, and chordate collections. The absolute number of digitized data records presented in GBIF is comparable for each group. However, the proportion of all North American arthropod specimens that have a digitized record is less than 5%, whereas that proportion is 15% for plants and higher for chordates. Moreover, because the total number of estimated arthropod species in North America is much greater than chordates and plants combined, the average number of specimens digitized per arthropod species (n = 97) lags significantly behind both plants (n = 404) and especially chordates (n = 2,584).

In addition, GBIF currently serves some 330 million non-specimen-based records (e.g., eBIRD, Sullivan et al., 2009) and image-only records (e.g., iNaturalist, Nugent, 2018) for chordates, which is nearly two orders of magnitude more than for plants and arthropods. In this regard, we also note that the Botanical Information and Ecology Network (BIEN) holds over 100 million observational records for New World plants (Enquist et al., 2016). In contrast, North American arthropods are only recently gaining traction in this arena, primarily due to citizen science initiatives such as iNaturalist, BugGuide.net, and other efforts focused on Lepidoptera (e.g., ButterflyNet, Pollardbase) and Odonata (e.g., Xerces Society Dragonfly Pond Watch Project).

The grand digitization challenge for North American arthropod collections

Given that North American collections hold approximately 300 million specimens, on what timeframe can we expect there to be a digital record available for each of those specimens? Fig. 2 provides a visual representation of this “grand challenge.” Our analyses indicate that some 2 million new digitized records are being produced annually from specimen labels, but as promising as this ongoing rate may be for generating large amounts of biodiversity data, there are still more than 280 million specimens remaining to be digitized. Currently, we are not transcribing enough specimen labels to keep up with new specimen acquisitions. A four-fold increase in our transcription rates is needed to capture label data for most specimens by mid-century (2050), assuming a 1% annual growth rate in specimen holdings.

Figure 2 The grand challenge for North American arthropod collections.

(A). Number of records of specimens digitized through 2018 (blue bar, in millions) and the total number of specimens in collections (green bar). (B) Projections of ongoing acquisition rates for specimens, compared to rates of digitization.

The majority of the 223 collections and 300 million specimens in North America are located in the United States, although Canada and Mexico have representative holdings for their respective countries (Fig. 3, Table S1). Canada has at least 17 collections and 32 million specimens, with the Canadian National Collection in Ottawa, Ontario including 17 million of those specimens. The National Autonomous University of Mexico (UNAM) houses three million arthropod specimens, and its holdings comprise 97% of all estimated Mexican specimens in the country (but only seven other major collections were identified in Mexico). There are no published estimates for the number of arthropod species occurring in Mexico. However, some data are available for select groups such as the Arctiini (Lepidoptera: Noctuoidea: Erebidae: Arctiinae). In the United States and Canada, there are 237 species described in this tribe (Lafontaine & Schmidt, 2010) but over 385 species occur in Mexico (Diaz, 1996), which represents a 62% greater species diversity in Mexico. If co-occurring species are removed, about twice as many Arctiini occur in Mexico (n = 289) compared to the United States and Canada (n = 141). These estimates are similar to a recent study demonstrating that vascular plant diversity is approximately 49% greater in Mexico compared to Canada and the United States (Ulloa et al., 2017; despite the fact that Mexico contains only about 10% of the land area of Canada and the United States combined).

Figure 3 Number of arthropod collections (blue) and number of specimens (green) for North American collections.

The current percent of specimens whose label data have been transcribed is above each bar.

Given its greater projected arthropod diversity, Mexico would need to increase its specimen holdings 60-fold to generate a corpus of specimens comparable to that of collections in the United States and Canada. In terms of digitization progress, Mexico has conducted a major effort via CONABIO that resulted in 33% of their existing specimen labels being transcribed. This is a much greater proportion than either Canada (3%) or the United States (6%) has achieved to date.

The ADBC initiative

Historically, individual taxonomists or ecologists working on a specific arthropod species and/or region conducted most digitization efforts, and those data were rarely shared. In just the past decade, the entomological community has made great strides in digitizing specimens and sharing those results (Fig. 4). This effort has benefitted enormously from The National Science Foundation’s (NSF, 2019) Advancing Digitization of Biodiversity Collections (ADBC) program (Page et al., 2015). ADBC began in 2011 and runs through 2021. More broadly, ADBC is enhancing and expanding the national resource of digital data that documents biological and paleontological collections, and is advancing scientific knowledge by improving access to digitized information (Nelson & Ellis, 2018; Page et al., 2015). There are other ongoing and significant programs occurring around the world to support direct digitization and/or informatics, such as the Atlas of Living Australia (Belbin & Williams, 2016), SpeciesLink in Brazil (Candela et al., 2014) and the Distributed System of Scientific Collections (DiSSCo), which is a similar European effort to digitize natural history collections (Addink, Koureas & Casino, 2018).

Figure 4 Number of digitized occurrence records for arthropod specimens from North American collections.

Estimates before 2010 are from Miller (1991), estimates since are from periodic queries of GBIF and SCAN.

The ADBC program has also promoted the development of a strong national investment in curation of the physical objects in scientific collections, and it contributes vitally to scientific research and technology interests in the United States. For arthropods, the impact of the ADBC program has been transformational from its inception, with the number of publicly available records having grown exponentially. Direct ADBC funding for digitization has produced about six million digitized records, and ADBC has indirectly spurred other collections to digitize their holdings. The NSF Collections in Support of Biological Research (CSBR) program has also emphasized digitization in its more recently funded CSBR awards.

The ADBC program has funded five Thematic Collections Networks (TCN) based on extant arthropods: InvertNet, Tri-Trophic, SCAN, LepNet, and Terrestrial Parasite Tracker, with an additional TCN focused on invertebrates (InvertEbase) and an invasive species TCN that includes arthropods. The primary TCN emphasis is on capturing descriptive data from specimen labels. However, collections are beginning to generate other data, such as geography, environmental habitat, phenology, associated organisms, collector field notes, and tissues and molecular data from specimens, which represent a rich biodiversity resource.

To expand on the recent ADBC efforts, we categorized North American collections into three groups based on digitization effort: (1) digitization not yet initiated; (2) records contributed to iDigBio, but no active digitization program in place; or (3) records contributed to iDiBio and with an active digitization program (Fig. 5). We distinguished the latter two categories by whether there was an existing GBIF IPT (Integrated Publishing Toolkit) as an endpoint serving Darwin Core Archive data. It is encouraging that collections with active digitization programs account for 68% of the specimens in US collections, and that smaller collections that have not yet contributed data to public portals only account for 7% of collections. However, this underscores the need to extend digitization practices to smaller collections, because smaller collections are focal points for mentoring students who contribute to the national workforce. A major challenge will be sustaining activities begun by ADBC activities once funding for the program ceases in 2021, such that collections can continue to integrate digitization into their everyday workflows.

Figure 5 Number of US collections and percentage of US specimens.

Collections are arranged by degree of digitation effort; see text for elaboration of effort categories.

Collection holdings: are we meeting research data needs?

It has been 28 years since Miller (1991) conducted the first and only comprehensive review of the 26 largest entomological collections at the time in the United States and Canada. The Miller review emerged from a 1988 meeting of the Association of Systematics Collections (ASC) that sought to address the capacity of systematics collections to increase research productivity, and proposed where national resources should be invested. As a measure of sustainability, the 26 collections in the Miller study have shown a steady 1% annual growth in the number of specimens, and the relative ranks of the collections have likewise remained rather stable (Fig. 6, Table S2). We lack comparable statistics for the other 197 collections we surveyed in North America, but there are now 58 collections that house as many or more specimens in 2018 than the 26th largest collection did in 1991 (see Table S1). Entomology collections in North America generally appear to be growing in the last ∼30 years.

Figure 6 Growth in number of specimens at the 26 largest collections in Canada and the United States over three decades.

Estimates from 1980s tabulated by Miller (1991), 2018 estimates extracted from this review (Table S1).

Are we collecting enough specimens?

North American collections have continued to grow over the three decades since Miller (1991) published his seminal paper, but we still ask whether we are collecting enough specimen data to address present and future biodiversity research needs. It is challenging to secure sufficient resources to store and maintain specimens even with a steady but low 1% annual growth in specimen acquisition. Furthermore, it is becoming increasingly difficult to justify financial and personnel support for collections without making specimen data fully available to researchers and educators. With the exception of a few dedicated funding programs at NSF and the Institute of Museum and Library Services (IMLS), digitization has been a largely unfunded mandate for most institutions, adding significant budgetary pressure (Blagoderov et al., 2012; Heidorn, 2011; Poole, 2010). Despite these challenges global change impacts have elevated the urgency to develop regional to continental strategies for reaching appropriate targets in specimen holdings (Sánchez-Bayo & Wyckhuys, 2019).

Here we initiate a needed discussion to assess the adequacy of current and projected holdings. Are there enough arthropod specimens available now in collections for biodiversity-related research? Will a projected 1% annual increase in specimen holdings meet expected future data needs? We know that there are unmet research needs for specimen data (Kharouba et al., 2018), and we are years away from knowing the degree to which the existing 300 million arthropod specimens will meet biodiversity research needs. Our aim is to begin a discussion about what our goals should be in terms of providing enough data for biodiversity research in the absence of a complete digitized specimen database.

We present a range of projections in growth of North American holdings, the most conservative is the current rate of 1% per year accumulation (Fig. 7). We do not have the capacity to determine what the upper rate of accumulation should be in the absence of full digitization of the 300 million specimens. However, it is useful to compare efforts to digitize North American arthropods with that for vertebrates (see (Guralnick & Constable, 2010)). Table 1 indicates that the average number of specimens digitized per arthropod species is 97, compared to 2,584 for chordate species, a 26-fold difference. We suggest that arthropod collections aim high and seek to digitize 2,500 records per species to match efforts for chordates. We are not suggesting that 2,500 records are required for every arthropod species to address every question. Depending on the nature of the question, only a fraction of all available records may be appropriate either because they do not address the question or data quality issues (Piel, 2018; Veiga et al., 2017; Sikes et al., 2016; Ferro & Flick, 2015), such as data leakage with historic records (Peterson et al., 2018). Future analyses should provide more refined per species digitization targets (Lobo et al., 2018; Pelletier et al., 2018) once more digitized arthropod records become available.

Figure 7 Projected growth in specimen numbers that would be required to meet data demands for biodiversity research.

Values expressed as percent increase in North American holdings for all collections in North America. Red circles indicate goals under the two trajectories.

We predict that to have a comparable corpus of arthropod data relative to chordates for North America, collections would need more than 360 million specimens to address data needs (Fig. 7). This assumes that 60% (181 million) of the current 300 million specimens in arthropod collections are from North America, which may be an overestimate (but freshwater mollusk collections are estimated to be 60% for Canada and the United States; (Sierwald et al., 2018; Solem, 1975)). The current rate of new specimen acquisition is insufficient, and even a doubling of the existing rate means that the target of 360 million would not be achieved until 2050. That target would be reached in 2047 if the overall rate of specimen acquisition were increased by 2.5% per year, by 2042 if it were increased to 3% annually and by 2030 if it were increased by 6% per year (Fig. 7).

Two reasons to aim for 2,500 digitized records per arthropod species are taxonomic skew and spatial bias in digitized records. The average number of digitized records per North American arthropod species is 97 (Table 1). However, less than 15% of all 142,800 species have that many records, and only 0.1% have over 2,500 records. The most recorded species is Bombus bifarius (Cresson), a common bumblebee in western North America, with over 26,000 records. Even still, at its northern (Alaska) and southern (Arizona, Nevada and New Mexico) limits of this species’ range, large gaps are present where there are few or no data records in areas they likely occur. This underscores that data bias can occur for even heavily sampled species (Ruete, 2015). Moreover, many distribution maps for arthropod species (and other taxa) are incomplete and biased due to an overrepresentation of localities favored by collectors (e.g., roads, popular landmarks), in regions of otherwise more broadly suitable habitat. In addition to spatial bias, historical degradation of locality records is a major challenge (e.g., geopolitical name changes or imprecisely described localities; (Bartomeus et al., 2018)). One useful effort would be to resample for species that either have reliable historic records, and/or have the most vulnerable habitats that are either experiencing change or are predicted to change.

Assessing what is an adequate number of specimens has been initiated for two arthropod Thematic Collections Networks (SCAN, LepNet). Taxa being targeted range from individual species of conservation concern (e.g., Poweshiek Skipperling, Oarisma poweshiek (Parker);(Belitz et al., 2018) to all Puerto Rican Lepidoptera that are susceptible to hurricanes (Seltmann et al., 2017). In the case of O. poweshiek, it was determined that there were adequate numbers of existing specimens and observational records. For the assessment of Puerto Rican Lepidoptera, this prompted the launch of a longer-term inventory to obtain more complete collections of all Lepidoptera (N Cobb, Catherine Hulsof, pers. comm., 2019). It is possible to provide reasonable running estimates for most North American species to include basic metrics such as number of occurrences through time documented in suitable habitat or range. These can be used to guide individual species studies to target likely areas where species occur but have not been documented or resample historic areas to confirm their presence. The data for groups of species can be integrated into a more strategic plan to direct future sampling campaigns.

US collections by holding size

Published reviews of natural history collections have focused on the collections with the largest specimen holdings (Dunnum et al., 2017; Miller, 1991; Short, Dikow & Moreau, 2018; Sierwald et al., 2018; Singer, Love & Page, 2018). Here, we consider all collection sizes for the three North American countries, with a focus on the United States because it has more readily available data. We summarize basic characteristics of Tier 1 (largest) through Tier 4 (smallest) collections in the US, including the number of collections, number of specimens, the percentage of collections that have initiated digitization, and the percentage of specimens that have had their labels transcribed for collections that are digitizing (Fig. 8). As expected, most collections are smaller (Tiers 3–4) although the absolute number of specimens is concentrated in larger collections (Tier 1). Small collections may face challenges in initiating digitization, but once begun, they processed a far greater percentage of their holdings than large collections. This suggests that although NSF ADBC funding has been effective in promoting digitization across collections, it has not had as large of an impact on the largest collections, where most specimens are located.

Figure 8 Attributes of 189 US collections arranged by size.

Tier 1: <100,000 specimens, Tier 2: 100,000 to 1,000,000 specimens, Tier 3: 1,000,000 to 3,000,000 specimens, Tier 4: Over 3,000,000 specimens. Numbers within black bars either represent the numbers of collections (A, C) or percentage values for each Tier (B, D).

Table 2 shows additional metrics as a function of collection size. A general concern among thematic collection networks was whether smaller collections could adequately image specimens, provide digitized specimen data with species-level identifications, and properly georeference localities. However, we found relatively few significant differences in statistics among tiers, although smaller collections appeared more effective in imaging, and small to intermediate sized collections more effective in identifications and georeferencing. We expected larger collections to have more global taxonomic and geographic coverage. To assess this, we measured the percentages of (1) non-North American records, (2) number of countries or large regional areas or islands, (3) total number of species recorded, and (4) the average distance of specimens from the collection itself. We predicted that smaller collections would have a strong regional focus and so we quantified (5) the percentage of specimens taken within a 50 km radius of the collection as a metric for a regional focus, and (6) the average rank collecting for each collection within the 50 km radius. These metrics supported our expectations, underscoring a more global taxonomic and geographic focus with increasing collection size. Distance from collection indicated a decreasing regional focus from Tier 4 to Tier 1 collections, although all collections had significant regional representation. The closest collection was almost always ranked first for having specimens from within 50 km of the collection. The only discrepancies occurred when two or more collections were physically near each other (e.g., Essig Museum in Berkeley, CA and the California Academy of Sciences in San Francisco, CA), or in a few Tier 4 collections (e.g., San Diego University, CA) where holdings strongly reflected a curator’s research interest in taxa distributed outside of North America.

Table 2 Summaries of metrics for digitized records from the four tier categories based on collection size.

Standard error of means are provided where applicable.

	Small	Large		
Collection size categories	Tier 4 <0.1 million	Tier 3 0.1 to <1 million	Tier 2 1 to 3 million	Tier 1 >3 million	Trend	
Data Quality						
Georeferenced	60% (+11)	72% (+9)	72% (+8)	60% (+8)	none	
Identified to species	51% (+8)	62% (+6)	70% (+6)	57% (+7)	none	
Records with images	22% (+10)	19% (+8)	6% (+4)	11% (+6)	down	
Regional to Global Metrics						
Non-North America records	15% (+7)	10% (+3)	20% (+6)	48% (+9)	up	
# of Countries/major regions	69	61	197	355	up	
Species per collection	631 (+258)	2,713 (+437)	4,451 (+1,353)	16,990 (+6,884)	up	
Distance from Collection (km)	881 (+343)	621 (+146)	1,106 (+174)	2,850 (+725)	up	
% of records (50 km radius)	85 (±5)	63 (+5)	62 (+5)	43 (+7)	down	
Mean rank (50 km radius)	1 (±0.0)	1 (±0.0)	1.1 (+0.1)	1.5 (+0.2)	none	

Possibly the most important metric regarding digitization was the number of “historical” records, which we defined as specimens collected prior to 1965, because these specimens represent perhaps the only direct evidence for pre-global change impacts. Our results show that large collections had more “historical” records than smaller ones (Fig. 9), and that there are at least 32 million “historical” specimens in US collections that can be used to assess global change impacts on arthropods. This is encouraging but presents a challenge because specimens are typically not separated by sampling year in collections, and hence cannot be readily targeted for digitization. The typical practice for digitization is to digitize all specimens in a drawer, as it is extremely inefficient to digitize a fraction of specimens in a drawer or unit tray. Following Allan et al. (2019), we believe it is important to target special collections of historic importance and develop more effective ways to increase the overall efficiency of digitization.

Figure 9 Estimates for numbers of specimens collected prior to 1965 in US collections.

Tier 4 collections hold the vast majority of “historical” specimens.

Discussion

Moving forward: challenges and opportunities

Our review is the first to provide a modern comprehensive assessment of arthropod collections in North America, and examine trends in the acquisition of new specimens and digitization of existing specimens. Both are important to address national/global needs for biodiversity data, and promote collaborative networks among North American collections (organizations such as the Entomological Collections Network [ECN], CONABIO, and Canadensys already serve in this capacity and are well positioned to collaborate). Below we summarize key points of our findings, and propose actions needed to mobilize more collections-based arthropod data, to maintain the transformational effort initiated by the NSF ADBC program and link to more global efforts.

Increasing specimen holdings

We suggest that North American collections should consider targeting the highest rate of specimen increase above the current holdings of North American arthropod specimens by at least an additional 100 million specimens by 2045 to marshal sufficient data to address global change impacts at the species level. This projection is based on the fact that less than 5% of all arthropod specimens in collections and only 0.1% of all arthropod species in collections are represented by species that have 2,500 digitized records/species—the average number of specimen records/species digitized to date for North American chordate species. One hundred million specimens is a rough estimate that will have to be refined and gap analyses should be done at the species level for priority arthropod taxa as we increase digitized records from collections, and develop research coordination networks to help guide and prioritize future surveys and digitization.

If we use estimates required for species distribution models, the expected standard for adequacy is growing, especially for species that occur over environmental gradients (Araújo et al., 2019). Thus, the target number of 100 million arthropod specimens may be an underestimate, given that 40% of the records in US collections are for specimens outside North America. It is possible that the high-end target of 100 million more specimens could be an overestimate, and underscores the need for additional assessments across different arthropod groups. Additionally, well-planned surveys could provide more complete coverage with fewer specimens than suggested by the this target.

Increasing digitization efforts

We estimate that data label transcription rates will need to increase by at least four-fold if the rate of new specimen acquisition increases to 3% per year. This goal may be achievable if robotic technologies (e.g., Beyond the Box (Maglia, Whiteman & Gropp, 2019)) can be implemented at just Tier 1 collections. During the NSF ADBC funding years, a number of collections developed protocols for mass digitization of newly obtained material that are much more efficient than digitization of specimens already integrated into collections. Because Tier 3–4 collections only account for 6% of specimens in North American collections, they will not directly impact the total number of records, but they will have a significant effect on filling in regional gaps and/or focusing on specific arthropod taxa, and they are important for recruiting new biodiversity researchers.

Citizen science and computer-aided identification

To what degree can citizen science efforts help address the burgeoning arthropod data needs? Approximately 10% of arthropod species are thought to be identifiable to species using an image, date and geographic point location (Poremski & Cobb, 2019). As smartphone cameras improve, reference image databases expand, and citizen science programs like iNaturalist (Nugent, 2018) and Fieldguide (Seltmann et al., 2017) continue to grow, we expect this to motivate biodiversity researchers to consider utilizing field images to augment physical specimens. Images are currently accepted by GBIF as machine observations and along with human observations comprise the vast majority of GBIF records. The primary concern is that there is no physical specimen to confirm, and the vetting process is not as rigorous as desired. To date, records provided by iNaturalist to SCAN are primarily for those groups that are generally well known to entomologists. These include most species of Orthoptera, Odonata, and many Lepidoptera, along with specific taxa from other orders (e.g., Coccinellidae). Other arthropod orders (e.g., Araneae) still need to be evaluated to determine the degree to which species-level identifications can be obtained from images. Additionally, further genetic information on cryptic species (Miller et al., 2016) may identify more taxa that require more than images to obtain species-level identifications. Using images for identification will significantly help fill current gaps in arthropod data records, and occurrence records do not generally need to be transcribed from images (since modern phone cameras provide coordinate data). Heberling & Isaac (2018) list a suite of variables that can be captured by images of plants that are not typically available from herbarium specimens (e.g., color, biotic associations, habitat). The same is true for arthropods. All arthropods stored in alcohol or collected in ethyl acetate can experience color fading, and specimens left in sunlight or under fluorescent lighting can also lose their color. Host plant associations are typically not recorded, and if they are recorded, the plant specimen is usually not submitted as a corollary herbarium specimen. Computer-aided identification accuracy is increasing exponentially, with the primary limitation being the lack of training images for neural networks (Schuettpelz et al., 2017). Although data associated with specimens (images, genetics, observations) can help augment arthropod biodiversity data needs, they will never replace whole-specimen repositories.

Coordination among North American countries

Although Mexico has made the greatest strides in digitization progress (33% of their specimen labels are transcribed), the 3 million specimens in Mexican collections remains low given that there are likely over 50,000 arthropod species in Mexico. Unlike the US and Canada, there are significant Mexican specimen holdings in institutions located in countries outside of Mexico. Many US taxa extend into Mexico, but the available data records often stop at the border (see Fig. 10). There should be additional cross-country network development, (but note collaborative informal networks such as the Madrean Biodiversity Project that hosts various expeditions to northern Mexico (Gottfried et al., 2013)).

Figure 10 Heat maps showing distributions for Bembidion (Carabidae) and Lasius (Formicidae) from SCAN data.

The dashed ellipses show a “border impact” where there is strong coverage in the US but almost no records in Mexico. The genera Bembidion and Lasius are representative of most arthropod taxa. Record density ranges from red (high) to green (low). Data derived from SCAN Spatial Module (heat map radius = 1, blur = 4). Map data by OpenStreetMap, under ODbL.

Specimen holdings in Canadian collections are primarily of specimens from Canada and the northern US, and total around 32 million specimens. To date, Canada has recorded 20% of the species diversity of the US but northern Canada, which harbors unique ecological habitats, is facing destruction and the remainder of the country may likely experience dramatic ecosystem conversion. The focus on the Arctic constitutes one of NSF’s 10 Big Ideas for future research (NSF, 2019). This NSF program should provide impetus for more specific planning and increased coordination among North American collections. Collections-based research will be important to these efforts, and there should be a North American effort to conduct repeated surveys (e.g., on a 3–5 year basis) to document the expected changes in the north.

Developing a collections-based network

Data collected during this review provide the basis for a permanent online repository similar to the Index Herbariorum for plant collections (Thiers, 2015). We present a basic information framework in Table S1 necessary to establish such an online resource, in which each collection could maintain its own data and integrate information from future work. We encourage the development of an “Index Entomologica” which could progressively add content such as sustainability scores for each collection based on criteria already established by the Index Herbariorum. The Entomological Collections Network (ECN; (Miller, 1991)) acts as an umbrella organization for entomology collections to share best practices, and it could play a major role in supporting an Index Entomologica, along with other organizations such as the Society for the Preservation of Natural History Collections SPNHC (Zimkus & Cundiff, 2019). Although the ECN is primarily active in the United States, it also includes Canada and Mexico and is in a position to network further with entomology collections around the world. An Index Entomologica would be synergistic with the proposed “Extended Specimen Data” program that has emerged as the focus of future biodiversity efforts from the Biodiversity Collections Network (BCoN, (Gropp, 2018)). Given that at least 90 million specimens in US collections are from countries outside of North America, the timing is ripe for North American collections to help build a global network with collaborations including e.g., iDigBio, GBIF, DiSSCo (Addink, Koureas & Casino, 2018), and SpeciesLink (2014).

Next-generation collections

With a cohesive North American collection network in place, a new strategic plan should be implemented to augment the current rate of 1% annual growth in acquisition of new specimens. Identifying gaps in taxonomic and geographic representation will lead to prioritization for collecting campaigns (e.g., the New Arctic (Cordova, 2016)). Existing collecting campaigns can also expand their efforts through temporary curation of by-catch samples to be shared with other researchers. The community as a whole should digitize and share by-catch samples (already implemented as part of the NEON ground dwelling carabid project). We have already seen a similar community effort in digitization campaigns in the LepNet TCN, where a group of over 50 collections focused their efforts on 3 target families of Lepidoptera, representing some of the most charismatic within the order (Papilionidae, Saturniidae, and Sphingidae).

NextGen collections is a new concept that has recently emerged from a national BCoN meeting (see themes outlined by Schindel & Cook (2018)). The primary focus is to promote integrated collections that include cross-phyla collections linked to environmental data gathered by deployable sensors. Collections are prioritized to address important social needs such as disease agents and pests. We fully support the NextGen concept, although the arthropod community still remains focused on filling taxonomic and regional gaps before this next step can be considered. Collecting data on associated taxa for key groups (herbivores, parasitoids, parasites, pollinators) and micro-environment data for other groups (detritivores, omnivores) are priorities. The resulting digitized data sets would promote more sophisticated and targeted efforts to better integrate data from collecting events.

NextGen collection practices will continue to arise in museums. For example, standard vocabularies will be developed for associated data denoting species associations (Poelen, Simons & Mungall, 2014) and specimen traits, among others. It may not be feasible to employ robotic systems in all collections, but we can implement this technology through funding by programs that emerge from NSF’s 10 Big Ideas (Cordova, 2016). Of the 10 Big Ideas, “Understanding the Rules of Life: Predicting the Phenotype” is perhaps the most relevant because of the potential for coupling specimen-based research with targeted NextGen collections, and integration with ecological studies to understand how phenotypes evolve. Employing such techniques at just the 30 largest collections would allow the digitization of most specimens in North America in a shorter time than what we have estimated. Computer-aided identification tools can be deployed to help curators sort and identify specimens, and should be incorporated into country-level to global strategic planning.

Conclusions

There are three major challenges and needs that remain for North American arthropod collections: (1) deploying effective strategies to integrate more specimens into collections; (2) improving of digitization workflows; and (3) better identification of societal needs for collection-based biodiversity information and conservation. To meet these challenges, there must be a strong call for a combination of technological development, financial and institutional resources needed to increase the capacity for needed specimens, and a better understanding of arthropods and their diversity. Increasing regional to global representation of arthropods will bring collections-based research to the forefront of addressing human impacts on our planet’s biodiversity.

Supplemental Information

Table S1 List of arthropod collections in North America

We include basic information including location, size, digitization effort and website.

Click here for additional data file.

Table S2 Comparison of holding size for the 26 largest collections in Canada and the United States in 1986 (Miller, 1991) compared to holding size in 2018

We calculate average number of specimens added per year and the percentage increase.

Click here for additional data file.

We thank Hojun Song and Jim Wooley (Texas A&M University) for providing their lists of collections, Larry Page (iDigBio, University of Florida) for providing lists of contacts. Katja Seltmann (University of California, Santa Barbara) helped conduct the initial set of analyses addressing digitization. Jesús Romero Napoles provided information on additional collections in Mexico. Scott Miller (Smithsonian Institute) and Joe Cook (University of New Mexico) kindly provided a review of the manuscript. We also thank the collaborating institutions that share data on the SCAN-LepNet portal, who provided data for this study. Any opinions, findings, and conclusions or recommendations expressed in this material are those of the author(s) and do not necessarily reflect the views of the National Science Foundation.

Additional Information and Declarations

Competing Interests

Author Contributions

Data Availability

The authors declare there are no competing interests.

Neil S. Cobb conceived and designed the experiments, analyzed the data, contributed reagents/materials/analysis tools, prepared figures and/or tables, authored or reviewed drafts of the paper, approved the final draft.

Lawrence F. Gall, Jennifer M. Zaspel, Nicolas J. Dowdy and Akito Y. Kawahara conceived and designed the experiments, analyzed the data, authored or reviewed drafts of the paper, approved the final draft.

Lindsie M. McCabe analyzed the data, contributed reagents/materials/analysis tools, prepared figures and/or tables, authored or reviewed drafts of the paper, approved the final draft.

The following information was supplied regarding data availability:

The raw data is available at Zenodo: Cobb, Neil, & McCabe, Lindsie. (2018). SCAN Data Set —10.22.18. http://doi.org/10.5281/zenodo.3476811

The SCAN Data Set 10.22.18 was drawn from the SCAN database, https://scan-bugs.org/.

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
