# Peer review of "Assessment of North American arthropod collections: prospects and challenges for addressing biodiversity research"

_PeerJ, doi:10.7717/peerj.8086_

## Round 0.1 · original submission · Minor Revisions

Neil, I very much enjoyed reading this. As you will see, the two reviewers offer a series of productive suggestions and I would like you to address all substantive ones. I agree with Andrew that the writing is somewhat U.S.-centric for an international journal (e.g., the discussion around NSF's 10 Big Ideas and any recommendations regarding NSF strategic planning) and suggest adding context that can readily be followed by “overseas” readers. Please explain acronyms such as SCAN when first mentioned. Reviewer 2 suggests further discussion of two points: (1) number of specimens to collect and digitize per species, and (2) digitizing level (the latter also addressed in Andrew's review) and I would find that useful as well.

·

Basic reporting

This manuscript provides a timely and useful analysis of the current state of entomological collections, in terms of digitization activity, data accessibility, and growth trends. A report like this is especially relevant, given relatively recent investments by funding agencies in the U.S. (especially the National Science Foundation) and elsewhere. The authors cite recent and important papers, provide relevant context and background information, and provide a number of charts and figures (maybe too many; could some be made supplementary?).

Experimental design

The authors' stated aims are to "[characterize] different types of arthropod collections, [articulate] challenges specific to arthropod collections, and [assess] digitization efforts to date with a focus on meeting research data needs". The primary author is in a prime position to perform this kind of analysis as one of the leading researchers of SCAN, the primary data portal through which most North American insect data are shared. As I stated earlier, this type of analysis is timely and useful, given that we are several years into a massive effort to digitize natural history objects and make the data available for research. I find no major flaws with their experimental design, and I appreciate the comparisons with collections of other taxa (plants, vertebrates) and between countries. I found the classification of collections into tiers to be unintuitive, mainly in that Tier 1 included the _smallest_ collections. Also, albeit a minor quibble, their criteria for what it means to be "digitized" are clear and relevant but leave me feeling unsatisfied. For example, we've invested *heavily* in terms of resources, training, person hours, etc. in digitizing our collection, but because we cannot satisfy all four criteria, especially "(3) capturing habitus image(s)" and (4) vetting species-level identifications" it would be difficult to say we digitized much. Maybe 1,000 of our specimens satisfy all 4 criteria?

Validity of the findings

Their results are useful if largely unsurprising. The authors provide a reasonable discussion of each result and a decent assessment of the impacts of the ADBC program. Their conclusions - backed by data - should help the collections community validate increased collection efforts and argue for increased investment in digitization, infrastructure, etc.

The discussion around NSF's 10 Big Ideas and any recommendations regarding NSF strategic planning are likely irrelevant to readers outside of the U.S. Generalizing these recommendations to "science-funding agencies" or some such would be more inclusive and timeless.

Additional comments

Altogether this manuscript represents a much needed analysis of the current state of the art in entomological collections in North America. You have interesting observations, great recommendations about how we need to scale up our collections and activity, and overall it's an interesting paper. The references need a bit of work, as many are malformed or are missing critical information. The English is clear but would benefit from some editing - remove extraneous commas, remove some redundancy, define acronyms when first used, add missing references, etc. I highlight these issues directly in the PDF.

Reviewer 2 ·

Basic reporting

Meets standards, including citing the more relevant prior literature, except that nearly a dozen of the citations in the text or Table 1 are not cited in References. Also, although all of the References are cited in the text, but many of them lack full publication data. These omissions are all highlighted and commented in the attached copy of the main manuscript.

Experimental design

No comment

Validity of the findings

No comment here, but see general comments below

Additional comments

This important work is a very welcome contribution, and should stimulate discussion and iimproved solutions for making collection data available to the wider community of naturalists. But the minor details are often sloppy (poor grammar, incomplete reference data or missing references, etc.) as noted in detail in the attached annotated pdf, and attention should be paid to fixing these details.

In general terms, the data sources are massive and comprehensive, and the analyses of these data and arguments drawn from them seem to be thoughtful and appropriate. But one of the most important resulting arguments (how many specimens to collect and digitize per species) seems to be one of the most weakly supported. I.e., the number the authors came up with for the proposed effort (2500 records for each species) was arrived at simply by aiming to “match efforts for chordates” (line 340), rather than based on some more objective calculation (or estimation) of how much information can be extracted from X number of specimens, and how much is enough to be useful. Such a goal (matching chordate information) is fine as an objective but probably quite unrealistic, given the far greater manpower and resources (on a per-species basis) available for chordate studies compared to arthropod studies, and the more charismatic nature of chordates which can generate public support and involvement more easily. Perhaps the authors could discuss this issue more, and work to strengthen their arguments for the number they came up with.

Lastly, one general limitation of the authors’ approach is that it refers only to collection digitization in the strictest sense, i.e. digitizing full individual specimen data as a mininum starting point. Although this is the ideal and desired endpoint, in practice the recommendations for collection growth and especially greater digitization effort are likely to be more aspirational than realistic in the several-decade time frame. More intermediate or coarse levels of digitization, such as collection inventories that record collection holdings (number of individuals of each species or unidentified lot and the region(s) they are from, and presence of types) can be prepared with 1-2 orders of magnitude less effort, and still provide invaluable online information to systematists (e.g., to determine if a loan request or collection visit is necessary in connection with an ongoing taxonomic revision) and also provide firm data about the current status of a collection to collection curators or managers to aid the preparation of grant proposals to complete the detailed digitization that is desired in the long term. Perhaps the authors could add a brief statement about this option as an intermidiate step?

Annotated reviews are not available for download in order to protect the identity of reviewers who chose to remain anonymous.

---

## Round 0.2 · accepted · Accept

Dear Neil, Thanks for the thorough responses to the reviewers' suggestions. I look forward to seeing this in "print" soon. Best, Rüdiger